# Genomic regions and candidate genes associated with seed nitrogen, phosphorus, and sulfur accumulation identified in the soybean 'Forrest' by 'Williams 82' RIL population

Nacer Bellaloui[1]*, Jiazheng Yuan[2], Dounya Knizia[3,4], Qijian Song[5], Frances Betts[2], Teresa Register[2], Earl Williams[2], Naoufal Lakhssassi[3], Hamid Mazouz[4], Henry T. Nguyen[6], Khalid Meksem[3], Alemu Mengistu[7], My Abdelmajid Kassem[2]

**1** USDA, Agriculture Research Service, Crop Genetics Research Unit, Stoneville, California, United States of America, **2** Plant Genomics and Biotechnology Laboratory, Department of Biological and Forensic Sciences, Fayetteville State University, Fayetteville, North Carolina, United States of America, **3** Department of Plant, Soil, and Agricultural Systems, Southern Illinois University, Carbondale, Illinois, United States of America, **4** Laboratoire de Biotechnologies & Valorisation des Bio-Ressources (BioVar), Department de Biology, Faculté des Sciences, Université Moulay Ismail, Meknès, Morocco, **5** Soybean Genomics and Improvement Laboratory, USDA-ARS, Beltsville, Maryland, United States of America, **6** Division of Plant Science and Technology, University of Missouri, Columbia, Missouri, United States of America, **7** USDA, Agricultural Research Service, Crop Genetics Research Unit, Jackson, Tennessee, United States of America

* nacer.bellaloui@usda.gov

## Abstract

Nitrogen (N), phosphorus (P), and sulfur (S) are essential nutrients for plant health. Deficiencies in N, P, or S in plants lead to lower seed production and seed quality in grain crops, including soybean seed. Soybean seed is a source of protein, oil, essential amino acids, and minerals. These nutrients are essential for plant health, and maintaining N, P, and S levels in soybean seed is crucial for higher seed nutritional value and amino acids quality. There is limited information on genomic regions, candidate genes, and molecular markers associated with soybean seed N, P, and S. Two field experiments were carried out in two locations using a 'Forrest' × 'Williams 82' recombinant inbred lines (RIL) population. A 306 RIL population and 2075 SNP markers were used to create the genetic map. The results showed a wide range of N, P, and S concentrations in both locations among RIL population lines. Based on the broad-sense heritability ($H^2$), 91.7% of seed N concentration variation was due to genetic effects, followed by 48.2% for S seed concentration, and a heritability of close to zero for seed P concentration. Eleven QTL were identified for seed N, seven QTL for seed P, and nine QTL for seed S in two locations. All these QTL had a significant linkage to the trait as their LOD ranged from 2.5 to 6.48 in 2018 and from 2.75 to 128.72 in 2020. Two QTL for seed N (*qN-02*-[IL-2020] on Chr 4, and *qN-03*-[IL-2020] on Chr 4 were identified at the marker Gm04_4687302-Gm04_7672403 and Gm04_7672403, and their LOD were 45.06 and 96.98, and their contribution to

**Data availability statement:** All relevant data are within the manuscript and its Supporting Information files.

**Funding:** This research was funded by the U.S. Department of Agriculture, Agricultural Research Service Project 6066-21220-016-000D, SIUC, UM, and FSU. The funders had no role in study design, data collection and analysis, decision to publish, or preparation of the manuscript.

**Competing interests:** The authors have declared that no competing interests exist.

the phenotypic variation were 45.85% and 48.37%, respectively. The low heritability of P indicated a major interactions between the trait (P) and environment. Except for the seed N, P, and S QTL, identified on Chr 16, 11 QTL reported here were not previously identified and therefore are novel. Several functional genes encoding N-, P-, and S-proteins, enzymes, and transporters were identified and located within the QTL interval. To our knowledge, the QTL identified here on Chr 2 and 6 are novel and were not previously identified. Therefore, QTL, genes, and molecular markers discovered in this research will provide breeders with new knowledge and tools for soybean selection for optimum seed mineral nutritional qualities. Also, this new findings advance our knowledge of physiology and genetics of seed N, S, and P candidate genes for genetic engineering application.

## Introduction

Soybean (*Glycine max* L.) is a major source of protein, amino acids, oil, fatty acids, and macronutrients, including nitrogen (N), phosphorus (P), and sulfur (S). Nitrogen, P, and S are essential nutrients affecting the growth, development, and crop production. Therefore, maintaining the optimum level of these nutrients in crop seeds is critical to ensure seed health. Nitrogen is essential for productivity [1–3], a source of amino acid and protein synthesis, and protein storage [4–9], DNA and RNA, phytohormones, co-enzymes, and involved in several physiological, metabolic, and biochemical reaction [10,11]. Phosphorus is a source of phytic acid (anti-nutritional component, especially at high level in seeds), and involved in cell membrane structure, function, cell membrane, lipid synthesis, ATP and NADP-H, DNA and RNA, carbohydrate metabolism, and active uptake [10–12]. Like N and P, S is also a critical nutrient and involved in specific amino acids such as S-containing amino acids (cysteine and methionine, two amino acids that are deficient in soybean cultivars, and as a result in protein meal) [13], enzymes and co-enzymes, and essential for N uptake [10–12].

Our literature search showed that genomic regions and gene candidates controlling seed N, P, and S accumulation is limited. Searching SoyBase revealed limited molecular markers and QTL associated with root and shoot N, P, and S, were identified, and not in mature soybean seed. (https://www.soybase.org/search/index.php?searchterm=Nitrogen±and±Phosphorus±and±Sulfur&list=bi_parental_qtl_list-view) [14]. For example, literature available reported QTL and molecular markers associated with shoot tissue concentrations of macro- and micro-nutrients, including N, P, and S [15]. They were able, in Genome-wide association (GWAS) studies using 31,748 SNPs, to identify several putative loci for macro-and micro-nutrients in soybean shoot, including one QTL for P (*qPHO001*), one QTL for N (*qNIT001*), one QTL for S (*qSUL001*). They suggested that QTL clustering of P, K, Mg, C, N, and S indicated physiological and genetic relationships, and possible similar metabolic processes between these nutrients. A QTL analysis was conducted to understand the genetic mechanism of leaf-related traits [15]. They used RILs of 200 individuals resulted from a cross between the cultivars 'Nandou 12' and 'Jiuyuehuang'. Liu et

al. (2019) found 6,366 SNPs markers that covered the whole genome of soybean distributed on 20 chromosomes and spanned 2818.67cM with an average interval of 0.44cM between adjacent markers. They were able to identify 19 QTL and 3 candidate genes associated with leaf-related traits [15].

Further, previous research indicated that most of the genetic mapping for plant nutrition was conducted on leaves, roots, or shoot [3,15,16], and not on nutrients N, S, and P accumulation in mature seeds [3,17,18]. Others were able to associate genotypic differences in soybean seed nitrogen accumulation with genomic regions controlling nitrogen accumulation in soybean during R5, R6, and R7 growth stages [19], but not at R8 (complete seed maturity stage) as in our current research. Panthee et al. (2004) used a population of 101 $F_{6:8}$ (F6-derived) recombinant inbred lines resulted from a cross between N87-984–16 × TN93–99. They found several QTL on chr 2 (D1b), 7 (M), 8 (A2), 14 (B2), 15 (E), 18 (G). They found that the phenotypic variation explained 5 to 11.6%. Also, others [16], working on 184 recombinant inbred lines resulted from Kefeng No. 1 and Nanong 1138–2 soybean varieties, were able to identify QTL associated with P deficiency tolerance in leaves, roots, and shoots. Li et al. (2005) identified seven QTL mapped on two chromosomes associated with weight of fresh shoot, P contents in leaf and in root. Other QTL associated with P in soybean shoot were reported elsewhere [20–22]. Further, using 92 $F_{5:7}$ (F5-derived) soybean RILs, derived from a cross between MD 96–5722 and Spencer using 5,376 SNP markers, they were able to identify QTL related to seed N, P, and S [23]. A QTL analysis was performed to understand the genetic mechanism of leaf-related traits [15]. They used RILs of 200 individuals resulted from a cross between the cultivars 'Nandou 12' and 'Jiuyuehuang'. They found 6,366 SNPs markers that covered the whole genome of soybean distributed on 20 chromosomes and spanned 2818.67cM with an average interval of 0.44cM between adjacent markers. They were able to identify 19 QTL and 3 gene candidates associated to leaf-related traits [15].

Based on the above literature, very limited information is available on soybean seed N, P, and S accumulation QTL. Therefore, the current research was aimed at identifying QTL associated genes for seed N, P, and S accumulation in a RIL population (total of 306 lines), using 5405 SNPs markers.

## Materials and methods

### Plant material and growth conditions

A RIL population ('Forrest' × 'Williams 82') was developed for genetic mapping [24]. 'Forrest' (PI 548402) [25] was created from a cross between 'Dyer' and 'Bragg', developed by USDA [25], and 'Williams 82' (PI 518671) [26] from a cross of 'Williams' and 'Kingwa' [26] (Fig 1). The genetic map was based on 306 RILs and 2075 SNP markers [27,28]. Our previous published research mapped QTL for soybean seed protein, oil, isoflavone, and amino acids, and showed that the parents (Williams 82 and Forrest) were contrasting in seed protein, oil, isoflavones, and some amino acids [28–30].

'Forrest' was originally developed for resistance to soybean cyst nematode (*Heterodera glycines*) and has been widely utilized in genetic mapping studies due to its agronomic resilience and seed composition traits. 'Williams 82' (PI 518671) [26], a cultivar released for resistance to *Phytophthora sojae*, serves as the reference genome in soybean genomics.

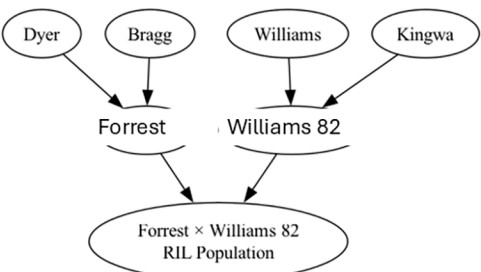

**Fig 1. Pedigree of the 'Forrest' × 'Williams 82' recombinant inbred line (RIL) soybean population used in this study.**

'Forrest' was released in 1973 by the USDA-ARS and Mississippi Agricultural and Forestry Experiment Station in cooperation with the Tennessee Agricultural Experiment Station [25]. It is a mid-maturity group V cultivar with a determinate growth habit, purple flower color, and yellow seeds. It has shown good performance under disease pressure and southern U.S. growing conditions. Forrest has been widely used in QTL studies due to its resistance traits and unique seed composition characteristics. 'Williams 82' (PI 518671), released in 1988 by the USDA-ARS and Illinois Agricultural Experiment Station [26], is a maturity group III cultivar with a determinate growth habit, purple flowers, and yellow seeds. It is used as the reference genome in soybean genomics due to its genetic stability and broad adaptability. Our previous published research mapped QTL for soybean seed protein, oil, isoflavone, amino acids, and showed that the parents (Williams 82 and Forrest) were contrasting in seed protein, oil, isoflavones, and some amino acids (included threonine, serine, proline, glycine, alanine, cysteine, valine, methionine, phenylalanine, lysine, tryptophan), as reported elsewhere [28–30].

While reports specifically comparing seed composition between these two cultivars are limited, our previous work has identified significant genetic differences in traits such as seed isoflavones [27], protein, oil, sugars, and tocopherols [29], and mineral nutrients [28,30]. These findings support the use of this RIL population for studying seed nutritional traits. The experiments were conducted on university research farms as described in Knizia et al. (2021) [27]. Two field experiments were conducted using the 'Forrest' × 'Williams 82' RIL population. The first experiment was carried out in 2018 at a Farm in Spring Lake, NC (35.17° N, 78.97° W) rented by Fayetteville State University. The second experiment was conducted in 2020 at the Southern Illinois University Agronomy Research Center in Carbondale, IL (37° N, 89° W). Both sites are managed by the university research farms with standard agronomic practices and irrigation capabilities. Experimental protocols, field layout, and management practices followed those detailed in Knizia et al. (2021) [27].

Seeds were planted on a 75 cm row-spacing, and the growth conditions and field management were conducted as previously reported by others [27,28]. In the current experiment, RILs showed a wide range of N, S, and P, and some RILs showed even higher concentrations of N, S, and P than the two parents. This means that RIL population showed Transgressive Segregation where the progeny shows more genetic variation and variation in gene expression than their parents. Also, this indicated that N, S, and P content trait in soybean is a complex polygenic trait, as previously reported [29], where QTL mapping for different seed nutrients, including N, S, and P using various mapping biparental populations. Therefore, QTL associated with N, S, and P were pursued.

### Analysis for seed N and S

The concentrations of N and S in the matured seed were determined by using ground and dried seeds using a Laboratory Mill 3600 (Perten, Springfield, IL USA). Nitrogen and S concentrations measurements in mature seeds were conducted using a 0.25 g ground-dried sample, and were combusted in an oxygen atmosphere at 1350 ºC, converting elemental N and S into $N_2$ and $SO_2$, respectively. Gases of $N_2$ and $SO_2$ were then passed through infrared cells and N and S were measured by an elemental analyzer using thermal conductivity cells (LECOCNS-2000 elemental analyzer, LECO Corporation, St. Joseph, MI USA) as previously detailed [31,32].

### Phosphorus measurement

Mature seeds at R8 were collected and analyzed for P concentrations using spectrophotometer. The concentration of P was determined by the yellow phosphor-vanado-molybdate complex as detailed elsewhere [33]. In short, a sample of 2 g of a dried ground seed was ashed, and then, 10 ml of 6 M HCl was added and placed in a water bath to bring the solution to dryness. Then, 2 ml of 36% v/v HCl were added, and the sample was boiled. The samples were boiled again by adding 10 ml of distilled water. The sample, then was transferred to a 50-mL volumetric flask, diluted to 50 mL with distilled water, filtered, and a volume of 2 ml of filtrate was first discarded and the rest was kept for P analysis. A volume of 5 ml of 5 M HCl and 5 ml of ammonium molybdate–ammonium metavanadate reagent were added to the filtrate, and the mixture was

diluted with distilled water to 50 ml. Ammonium molybdate–ammonium metavanadate was prepared by dissolving 25 g of ammonium molybdate and 1.25 g of ammonium metavanadate in 500 ml of distilled water. Dihydrogen orthophosphates was used to prepare a P standard curve (0–50 μg/ml of P). A Beckman Coulter DU 800 spectrophotometer was used to measure the P concentrations at 400 nm.

### DNA isolation, SNP genotyping, and genetic map construction

Genomic DNA of the RIL population and their parents was extracted according to others and our previous published research [27,28,30,34,35]. The RIL population was genotyped with BARCSoySNP6K Illumina Infinium BeadChips [35, https://www.soybase.org/tools/snp50k/]. Genotyping was conducted in the Soybean Genomics and Improvement Laboratory, USDA-ARS, Beltsville, MD, USA. A threshold of 2.5 for LOD, and a maximum genetic distance of 50 cM to group markers were used. The linkage groups were assigned to corresponding soybean chromosomes as described in SoyBase [36,37] (https://www.soybase.org/about/lgs_and_chromosomes/).

### N, S, and P QTL detection, candidate genes, and statistics

Detection of QTL and statistical analysis were performed as previously described [27,28]; the broad sense heritability ($H^2$) analysis from two-way ANOVA was performed using the equation:

$$H^2 = \text{sigma G2}/[\text{sigma G2} + (\text{sigma GE2}/\text{e}) + (\text{sigma e2}/\text{re})] \tag{1}$$

As described in details by others and by our previous published work [27,28,30,38]. The significance level was conducted using R package car (type II Wald chi-square tests) (R Software, accessed on June 15, 2023) [39]. To identify QTL for seed N, S, and P concentrations in RIL population, we used Composite Interval Mapping (CIM) methods of Win-QTL Cartographer 2.5 (Windows QTL Cartographer 2.5. Department of Statistics, North Carolina State University, Raleigh, NC; http://statgen.ncsu.edu/qtlcart/WQTLCart.htm) [40]. The default parameters of WinQTL Cartographer were selected (Model 6, 1 cM step size, 10 cM window size, 5 control markers, and 1,000 permutations threshold) [40]. Chromosomes were drawn using MapChart 2.2 [41]. Designation of QTL in the two years were done following the approach used by others [28,41]. We used CIM method because it is a widely accepted and robust method for detecting QTL with moderate to high power and resolution. We acknowledge that alternative methods such as Multiple QTL Mapping (MQM)or Inclusive Composite Interval Mapping (ICIM) may offer complementary insights, particularly in separating closely linked QTL or modeling epistatic interactions. However, the current study focused exclusively on CIM, and we did not perform MQM analyses. The candidate genes, within identified QTL for N, P, and S content in soybean seeds, were annotated using SoyBase Genome Browser (glyma.Wm82.gnm4).

A randomized complete block design (RCBD) was used for these experiments, with three replicates per genotype. This experimental layout was adapted from Knizia et al. [27], and followed standard field protocols for soybean QTL mapping studies. Since Hurricane Florence caused damage to the experimental site in Spring Lake, NC, in 2018, phenotypic and QTL analysis for that location was limited to 187 undamaged RILs (n = 187). However, the 2020 experiment in Carbondale, IL, was not affected, and data were collected from the full population of 306 RILs (n = 306). Analysis of Means procedure was conducted using Proc Means in SAS. Correlations were conducted by SAS using PROC REG (SAS, Statistical Analysis Systems, Cary, NC, USA, 2002–2012) [42].

### Results

A wide range of N, P, and S concentrations were observed in both locations (Table 1; Figs 2 and 3) due to environmental factors, including rainfall, wind, temperature. For example, Hurricane Florence occurred in Spring Lake, NC, in 2018, and this could be a source of variance, in addition to the genotypic variance among the lines in each year. Analysis of variance

**Table 1. Statistical components of soybean seed N (mg/kg), P (mg/kg), and S (mg/kg) in 'Forrest' by 'Williams 82' recombinant inbred soybean lines (RILs) population in 2018 NC and 2020 in IL.**

| Trait | Mean | Minimum | Maximum | Median | CV (%) | SE | Skewness | Kurtosis |
|---|---|---|---|---|---|---|---|---|
| | | | 2018[a] | | | | | |
| Nitrogen (mg/kg) | 63.3 | 54.7 | 69.4 | 63.4 | 3.91 | 0.17 | −0.36 | 3.51 |
| Phosphorus (mg/kg) | 6.2 | 4.8 | 7.84 | 6.30 | 9.77 | 0.04 | −0.19 | 2.19 |
| Sulfur (mg/kg) | 2.7 | 1.95 | 3.57 | 2.74 | 10.14 | 0.02 | −0.01 | 2.3 |
| | | | 2020[b] | | | | | |
| Trait | Mean | Minimum | Maximum | Median | CV (%) | SE | Skewness | Kurtosis |
| Nitrogen (mg/kg) | 62.57 | 52.6 | 94.5 | 59.10 | 6.94 | 0.07 | 1.73 | 4.34 |
| Phosphorus (mg/kg) | 5.96 | 4.26 | 7.74 | 5.93 | 8.17 | 0.03 | 0.27 | 3.89 |
| Sulfur (mg/kg) | 3.25 | 2.57 | 4.15 | 3.22 | 7.61 | 0.01 | 0.4 | 3.29 |

[a]Nutrients concentration (mg/kg) in 2018 in Forrest: N = 61.60; P = 6.66; S = 2.76; in Williams 82: N = 58.70; P = 6.92; S = 2.81. [b]Nutrients concentration (mg/kg) in 2020 in Forrest: N = 85.70; P = 6.83; S = 3.59; in Williams 82: N = 89.80; P = 6.98; S = 3.76.

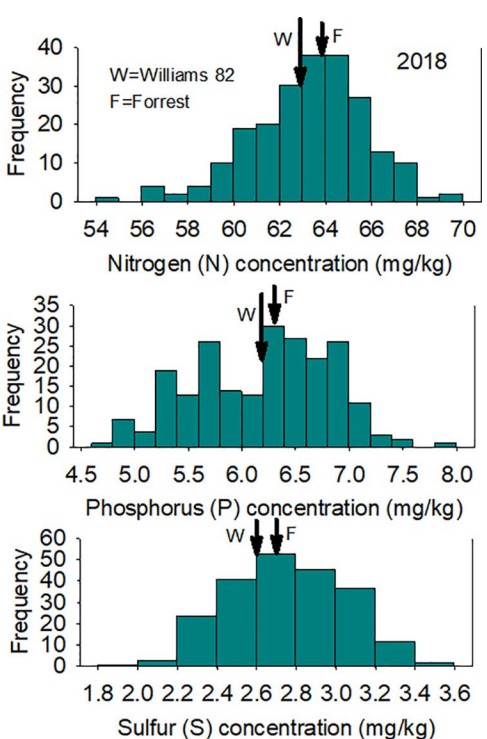

**Fig 2. Frequency distribution for seed N (Top), P (Middle), and S (Bottom) in 2018 in the 'Forrest' by 'Williams 82' recombinant inbred soybean lines (RILs) population in soybean.** Gaps that exist in any distribution graph indicate there is zero line in that range.

(ANOVA) showed that year had significant effects on N (F = 476.8; P = 0.0001), P (F = 32.8; P = 0.0001), and S (F = 386.7; P = 0.0001). Also, line had a significant effect on N (F = 29.3; P = 0.0001), S (F = 1.4; P = 0.008), but not significant for P (F = 0.9; P = 81). A 91.7% (H²) of seed N trait variation is due to genetics, followed by 48.2% for S seed concentration trait, and finally an inheritance of close to zero for seed P concentration trait (Table 2).

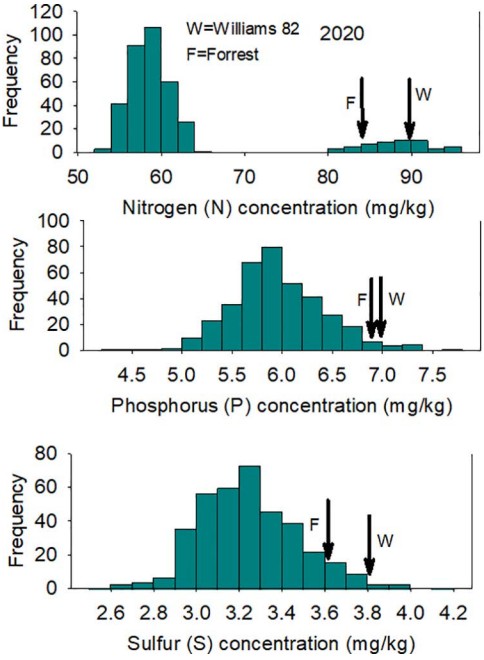

**Fig 3. Frequency distribution for seed N (top), P (middle), and S (bottom) in 2020 in the 'Forrest' by 'Williams 82' recombinant inbred soybean lines (RILs) population in soybean.** Gaps that exist in any distribution graph indicate there is zero line in that range.

**Table 2. Statistical components (Sum Square and Mean Square) and broad-sense heritability (H²) for soybean seed nutrients (mg/kg) N, P, and S in Forrest' by 'Williams 82' recombinant inbred soybean lines (RILs) population across two years (2018 and 2020).**

| Nitrogen | | | |
|---|---|---|---|
| | Sum Square | Mean Square | H² |
| Line | 35209 | 117.36 | 0.917 |
| Year | 1869 | 1869.44 | |
| Line:Year | 1762 | 9.68 | |
| **Phosphorus** | | | |
| | Sum Square | Mean Square | H² |
| Line | 85.934 | 0.2864 | -0.002 |
| Year | 6.616 | 6.616 | |
| Line:Year | 51.251 | 0.2871 | |
| **Sulfur** | | | |
| | Sum Square | Mean Square | H² |
| Line | 32.938 | 0.1098 | 0.482 |
| Year | 20.933 | 20.9334 | |
| Line:Year | 9.634 | 0.0529 | |

The low heritability of P indicated that almost all of the variability in this trait is due to environmental factors, and negligeable/very little effects is due to genetic differences. Except for the correlation between N and P in 2018, correlations between N, S, and P nutrients in two years and at both locations were all positive (Figs 4 and 5).

In 2018, the QTL for seed N explained 4.15% to 8.19% of the phenotypic variation; for seed P explained 4.15% to 11.10%; for seed S, the QTL explained 3.89% to 5.86%. In 2020, the QTL for seed N explained 7.14 to 78.52%; for seed P, the QTL

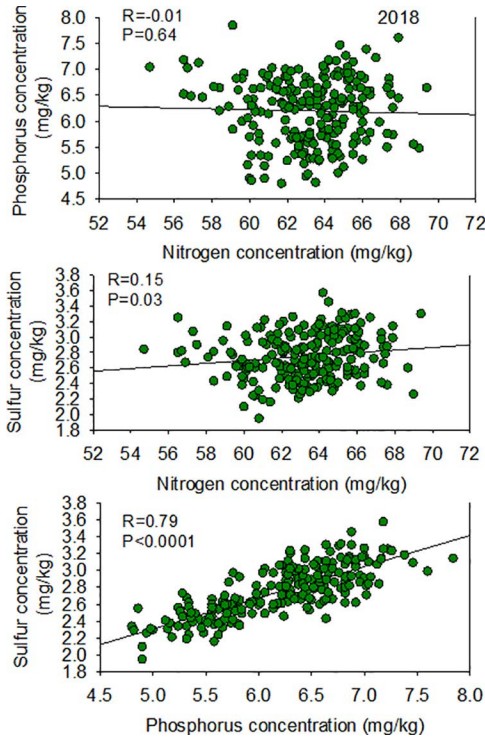

**Fig 4. Patterns of correlation between seed N vs. P (top), N vs. S (middle), and P vs. S (bottom) in 2018 in the 'Forrest' by 'Williams 82' recombinant inbred soybean lines (RILs) population in soybean.**

explained 4.26 to 12.17%; for seed S, the QTL explained 7.84 to 11.51% of phenotypic variation. All these QTL had a significant linkage to the trait as their LOD ranged from 2.5 to 6.48 in 2018 and from 2.75 to 128.72 in 2020 (Table 3; S1-S15 Figs).

A total of 11 QTL were detected for seed N on 9 chromosomes (Chr), seven QTL for seed P on 7 Chr, and 9 QTL for seed S on 7 Chr in two locations (Spring Lake, NC in 2018; and Carbondale, IL in 2020) (Table 3; S1-S15 Figs).

In 2018, and for seed N QTL, the highest LOD (4.51) was observed for N QTL *qN-01*-[NC-2018] on Chr 2 with the marker Gm02_5141136-Gm02_4938821 at marker intervals of 137.1-295.8 cM, and a contribution to the phenotypic variation of 8.19%. For P QTL, the highest LOD (6.48) was observed for QTL *qP-03*-[NC-2018] with the marker Gm12_975837-Gm12_1632399, at 178.7-189.3 cM, with a contribution of 11.10%. For S QTL, the highest LOD (3.57) was recoded for QTL *qS-01*-[NC-2018] on Chr 2 with a marker Gm02_9925870-Gm02_7987834 at 146.2-164.5 cM with a contribution of the trait of 5.86%.

In 2020, the LOD for seed N QTL was highly significant, but the highest (LOD of 128.72) were recorded for QTL *qN-01*-[IL-2020] on Chr 1 with the marker Gm01_3504836-Gm01_3466825 at 0.1-2.1 cM (Table 3). The second highest LOD was recoded for seed N QTL *qN-04*-[IL-2020] with LOD of 96.98 (Table 3; S1-S15 Figs). For seed P QTL, the highest LOD (9.42) was for QTL *qP-02*-[IL-2020] at marker Gm03_4469376-Gm03_4447541 and at the position 39.3-40.1 cM. For seed S QTL, the highest two LOD were recoded for QTL *qS-01*-[IL-2020] on Chr 2 with LOD 6.32, and at marker Gm02_1084314-Gm02_9925870 at position 138.6-144.2 cm; and for QTL *qS-06*-[IL-2020] on Chr 19 with LOD of 6.64 at marker Gm19_9978735-Gm19_3789399 and at position of 72.8-92.1cM. The contribution of the QTL *qN-01*-[NC-2018] on Chr 2, with the highest LOD (6.48), to the phenotypic variation was 8.19%; for seed P, the highest contribution (11.10%) to the phenotypic variation was recorded on QTL *qP-03*-[NC-2018] on Chr 12; for S, the highest contribution (3.57) to the phenotypic variation was observed on QTL *qS-06*-[IL-2020] on Chr 19.

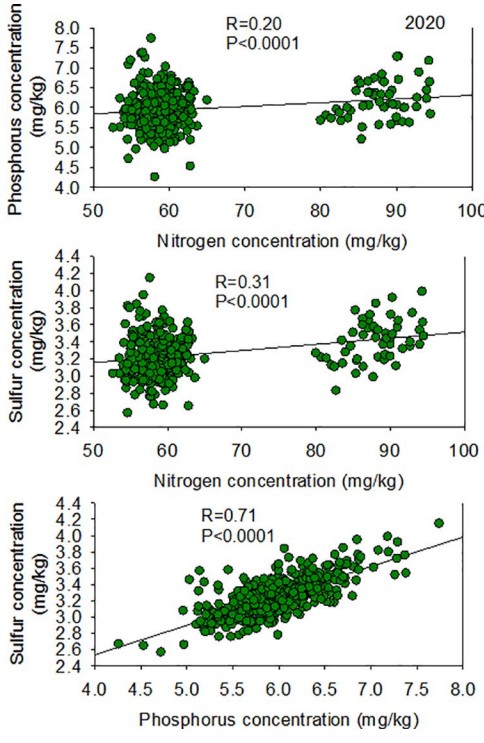

**Fig 5. Patterns of correlation between seed N vs. P (top), N vs. S (middle), and P vs. S (bottom) in 2020 in the 'Forrest' by 'Williams 82' recombinant inbred soybean lines (RILs) population in soybean.**

In 2020, and with the except for QTL *qN-07*-[IL-2020], the contribution of the rest of seed N QTL ranged from 37.57 to 78.52%, with the two highest were recorded on QTL *qN-01*-[IL-2020] on Chr 1with LOD of 128.72% with the marker Gm01_3504836-Gm01_3466825 at 0.1-2.1 cM; and QTL *qN-02*-[IL-2020] on Chr 4 with 96.98% with the marker Gm04_7672403 at position 16.5-28.5 cM. The highest contribution to the phenotypic variation for seed P QTL *qP-02*-[IL-2020] was recorded on QTL with 12.17%. For seed S, the highest contribution to the phenotypic variation was recorded for QTL *qS-02*-[IL-2020] on Chr 4 with 11.51%. In 2018, positive additive effects for seed N QTL and P seed QTL were observed; however, negative additive effects for S seed QTL (Table 3). A positive and negative additive effects for seed N QTL, P QTL and S QTL. It is clear the additive effect, either positive or negative were greater in 2020 (Table 3).

Two QTL for seed N (*qN-02*-[IL-2020] on Chr 4; and *qN-03*-[IL-2020] on Chr 4 were identified at the marker Gm04_4687302-Gm04_7672403 and Gm04_7672403, respectively, and respectively at peak of 1.4-5.4 cM and 16.5-28.5 cM. Their LOD were highly significant 45.06, 96.98 (Table 3; S16 Fig), and their contribution to the phenotypic variation were 45.85% and 48.37%, respectively for (qN-02-[IL-2020]. The additive effects were both positive and they were counted for 14.09 and 14.91, respectively for *qN-02*-[IL-2020] and *qN-03*-[IL-2020]. Similar observation was noticed for seed S trait for *qS-02*-[IL-2020] and *qS-03*-[IL-2020] on Chr 5 Gm04_4687302-Gm04_7672403 and Gm04_4388529-Gm04_4417048 at 1.4-5.4cM and 165.3-168.5 cM, respectively. Their LOD, respectively for *qS-02*-[IL-2020] and *qS-03*-[IL-2020], were 2.75 and 3.84 (both were significant), with a contribution to phenotypic variance of 11.51 and 4.5, and additive effects of 1 0.10 and 0.05, much lower than those of *qN-02*-[IL-2020] and *qN-03*-[IL-2020]. Using SoyBase Genome Browser (glyma.Wm82.gnm4), several functional genes encode N-, P-, and S-proteins, enzymes and transporters were identified and located within the QTL intervals (Tables 4 and 5).

**Table 3. QTLs that control seed nutrients (mg/kg) nitrogen (N), phosphorus (P), and sulfur (S) contents in two environments over two years (E1 and E2). E1: Spring Lake, NC (2018) and E2: Carbondale, IL (2020). Only QTL with LOD scores > 2.5 and identified by composite interval mapping (CIM) method of QTL Cartographer (Wang et al., 2012) are reported. Chr = chromosome; Position = The peak position of the significant QTL; LOD = Logarithm of the odds; $R^2$ = Percentage of variation explained by each identified QTL; Add. Eff. = Additive effect; a negative value indicates that the Forrest allele increased the trait value.**

### E1: Spring Lake, NC (2018)

| Trait | QTL | Chr. (Linkage group) | Marker | Pos. (cM) | LOD | $R^2$ | Add. Eff. |
|---|---|---|---|---|---|---|---|
| N | qN-01-[NC-2018] | 2 (D1b) | Gm02_5141136-Gm02_4938821 | 137.1-295.8 | 4.51 | 8.19 | 1.7601 |
| | qN-02-[NC-2018] | 7 (M) | Gm07_4292342 | 8.1-10.1 | 2.50 | 4.44 | 0.5294 |
| | qN-03-[NC-2018] | 15 (E) | Gm15_1142117-Gm15_1149627 | 140.2-140.8 | 2.72 | 4.54 | 0.5441 |
| | qN-04-[NC-2018] | 17 (D2) | Gm17_8449684 | 134.5-136.5 | 2.94 | 5.29 | 0.5958 |
| P | qP-01-[NC-2018] | 4 (C1) | Gm04_7672403 | 6.5-8.5 | 2.54 | 4.15 | 0.3114 |
| | qP-02-[NC-2018] | 5 (A1) | Gm05_1744708-Gm05_1705841 | 127.9-134.4 | 3.34 | 5.54 | 0.1805 |
| | qP-03-[NC-2018] | 12 (H) | Gm12_975837-Gm12_1632399 | 178.7-189.3 | 6.48 | 11.10 | 0.2527 |
| | qP-04-[NC-2018] | 13 (F) | Gm13_2748576 | 0.5-4.5 | 2.51 | 9.07 | 0.3739 |
| S | qS-01-[NC-2018] | 2 (D1b) | Gm02_9925870-Gm02_7987834 | 146.2-164.5 | 3.57 | 5.86 | −0.0815 |
| | qS-02-[NC-2018] | 5 (A1) | Gm05_3273418 | 33.1 | 2.50 | 3.89 | −0.1583 |
| | qS-03-[NC-2018] | 16 (J) | Gm16_3010888-Gm16_2998592 | 93.8-94.7 | 3.41 | 5.59 | −0.0667 |

### E2: Carbondale, IL (2020)

| Trait | QTL | Chr. | Marker | Pos. (cM) | LOD | $R^2$ | Add. Eff. |
|---|---|---|---|---|---|---|---|
| N | qN-01-[IL-2020] | 1(D1a) | Gm01_3504836-Gm01_3466825 | 0.1-2.1 | 128.72 | 78.52 | 14.73 |
| | qN-02-[IL-2020] | 4 (C1) | Gm04_4687302-Gm04_7672403 | 1.4-5.4 | 45.06 | 45.85 | 14.09 |
| | qN-03-[IL-2020] | 4 (C1) | Gm04_7672403 | 16.5-28.5 | 96.98 | 48.37 | 14.91 |
| | qN-04-[IL-2020] | 6 (C2) | Gm06_1737718-Gm06_5014399 | 30.7-58.4 | 25.53 | 37.57 | −11.13 |
| | qN-05-[IL-2020] | 10 (O) | Gm10_621706-Gm10_6020959 | 196.1-202.1 | 47.88 | 42.05 | −13.20 |
| | qN-06-[IL-2020] | 16 (J) | Gm16_7947809-Gm16_1079308 | 0.4-20.5 | 34.57 | 38.97 | −11.13 |
| | qN-07-[IL-2020] | 17 (D2) | Gm17_3709542-Gm17_5294475 | 0.7-1.1 | 7.56 | 7.14 | 12.04 |
| P | qP-01-[IL-2020] | 2 (D1b) | Gm02_5141136-Gm02_1020061 | 137.1-139.8 | 3.51 | 4.26 | −0.18 |
| | qP-02-[IL-2020] | 3 (N) | Gm03_4469376-Gm03_4447541 | 39.3-40.1 | 9.42 | 12.17 | 0.55 |
| | qP-03-[IL-2020] | 20 (I) | Gm20_480832-Gm20_1133712 | 141.5-156.1 | 4.12 | 5.34 | 0.13 |
| S | qS-01-[IL-2020] | 2 (D1b) | Gm02_1084314-Gm02_9925870 | 138.6-144.2 | 6.32 | 7.84 | −0.13 |
| | qS-02-[IL-2020] | 4 (C1) | Gm04_4687302-Gm04_7672403 | 1.4-5.4 | 2.75 | 11.51 | 0.10 |
| | qS-03-[IL-2020] | 4 (C1) | Gm04_4388529-Gm04_4417048 | 165.3-168.5 | 3.84 | 4.51 | 0.05 |
| | qS-04-[IL-2020] | 7 (M) | Gm07_1285811-Gm07_1045167 | 148.6-152.2 | 3.82 | 4.49 | 0.05 |
| | qS-05-[IL-2020] | 10 (O) | Gm10_4842697-Gm10_4777478 | 11.1-16.7 | 3.25 | 3.76 | 0.05 |
| | qS-06-[IL-2020] | 19 (L) | Gm19_9978735-Gm19_3789399 | 72.8-92.1 | 6.64 | 7.93 | 0.22 |

## Discussion

### QTL analysis for N, P, and S

The significant effect of line and location is due to the genotypic differences among lines and the contribution of the environment to the variation of the trait. The high heritability of 91.7% of seed N trait variation is due to genetics, followed by 48.2% for S seed concentration trait, and finally an inheritance of close to zero for seed P concentration trait. The very low inheritance, recorded in seed P may indicate the strong effect of environmental factors on the trait and very little effects due to genetic differences. Also, the low heritability that was shown in seed P could be due to complex quantitative trait, gene-to-gene interactions (epistasis effect), and significant gene by environment interactions with P, and suggest

**Table 4. QTL and candidate genes for seed nutrients (mg/kg) N, P, and S accumulation in the Forrest' by 'Williams 82' recombinant inbred soybean lines (RILs) population (RILs) population in soybean in two environments [(Spring Lake, NC (2018); and Carbondale, IL (2020)]. Only QTL with LOD scores ≥ 2.5 and identified by composite interval mapping (CIM) method of QTL Cartographer (Wang et al., 2012), were reported. Candidate genes, controlling N, P, and S accumulation associated with previously reported QTLs using SoyBase Genome Browser, are presented. Chr = Chromosome.**

| Trait | Environment | Chr (linkage group) | Genomic Interval | Candidate Genes |
|---|---|---|---|---|
| | qN-01-[NC-2018] | 2 (D1b) | Gm02_5141136-Gm02_4938821 | Glyma.02G043700/Glyma.02G041900 |
| | qN-02-[NC-2018] | 7 (M) | Gm07_4292342 Gm07_4292342 | Glyma.07G045900 |
| | qN-03-[NC-2018] | 15 (E) | Gm15_1142117-Gm15_1149627 | Glyma.15G015600 |
| N | qN-04-[NC-2018] | 17 (D1) | Gm17_8449684 | Glyma.17G110701/Glyma.17G106800 |
| | qN-01-[IL-2020] | 1(D1a) | Gm01_3504836-Gm01_3466825 | Glyma.01G033700/Glyma.01G033800/Glyma.01G033900/Glyma.01G034300 |
| | qN-02-[IL-2020] | 4 (C1) | Gm04_4687302-Gm04_7672403 | Glyma.04G058600/Glyma.04G060600/Glyma.04G073800 |
| | qN-03-[IL-2020] | 4 (C1) | Gm04_7672403 | Glyma.04G073800/Glyma.04G080700 |
| | qN-04-[IL-2020] | 6 (C2) | Gm06_1737718-Gm06_5014399 | Glyma.06G055100 |
| | qN-05-[IL-2020] | 10 (O) | Gm10_621706-Gm10_6020959 | Glyma.10G030800 |
| | qN-06-[IL-2020] | 16 (J) | Gm16_7947809-Gm16_1079308 | Glyma.16G041200 |
| | qN-07-[IL-2020] | 17 (D2) | Gm17_3709542-Gm17_5294475 | Glyma.17G067200/Glyma.17G067301 |
| | qP-01-[NC-2018] | 4 (C1) | Gm04_7672403 | Glyma.04G088000 |
| | qP-02-[NC-2018] | 5 (A1) | Gm05_1744708-Gm05_1705841 | Glyma.05G020500 |
| P | qP-03-[NC-2018] | 12 (H) | Gm12_975837-Gm12_1632399 | Glyma.12G014400/Glyma.12G019600 |
| | qP-04-[NC-2018] | 13 (F) | Gm13_2748576 | Glyma.13G004900 |
| | qP-01-[IL-2020] | 2 (D1b) | Gm02_5141136-Gm02_1020061 | Glyma.02G054800 |
| | qP-02-[IL-2020] | 3 (N) | Gm03_4469376-Gm03_4447541 | Glyma.03G043675 |
| | qP-03-[IL-2020] | 20 (I) | Gm20_480832-Gm20_1133712 | Glyma.20G011900/Glyma.20G018000 |
| | qS-01-[NC-2018] | 2 (D1b) | Gm02_9925870-Gm02_7987834 | Glyma.02G095500 |
| S | qS-02-[NC-2018] | 5 (A1) | Gm05_3273418 | Glyma.05G038100 |
| | qS-03-[NC-2018] | 16 (J) | Gm16_3010888-Gm16_2998592 | Glyma.16G030800 |
| | qS-01-[IL-2020] | 2 (D1b) | Gm02_1084314-Gm02_9925870 | Glyma.02G095500 |
| | qS-02-[IL-2020] | 4 (C1) | Gm04_4687302-Gm04_7672403 | Glyma.04G080300 |

limitations of these QTL use in the breeding selection [28]. It was reported on other minerals that the non-genetic factors could be extremely high [28,29]. They further added that the low heritability for P could be due to the interaction between the trait and environment. Also, the large interaction of gene with environment for P QTL may require a large number of RILs across locations and across years to obtain significant QTL with high inheritance [28,30]. This will require further research before final conclusions are made. The positive correlations between N, S, and P nutrients in both years and at both locations indicated the symport relationship between these nutrient transport system. The negative correlation between N and P in 2018 indicated antiport relationship that can occur due to environmental factors, especially drought, heat, and soil conditions. Positive and negative mineral relations were previously observed [11,12,43–45].

To our knowledge previous genomic research, conducted on minerals, was mostly conducted on leaves, roots, or shoot [3,15,16], and not on seed N, S, and P nutrients accumulation in soybean seeds [3,17,18]. For example, searching SoyBase (https://www.soybase.org/search/index.php?searchterm=Nitrogen±and±Phosphorus±and±Sulfur&list=bi_parental_qtl_listview) revealed that only QTL and molecular markers related to N, P, and S concentrations in shoot tissue were identified [17]. They were able to identify soybean shoot QTL (one QTL for P (qPHO001), one QTL for N (qNIT001), and one QTL for S (qSUL001). They explained that QTL clustering of P, K, Mg, C, N, and S indicated physiological and genetic relationships, and possible similar metabolic processes between these nutrients. Other researchers identified QTL related

**Table 5. Candidate genes and their functional annotation for seed nutrients ((mg/kg)) N, P, and S accumulation in the Forrest' by 'Williams 82' recombinant inbred soybean lines (RILs) population in soybean in two environments [(Spring Lake, NC (2018); and Carbondale, IL (2020)]. Only QTL with LOD scores ≥ 2.5 and identified by composite interval mapping (CIM) method of QTL Cartographer (Wang et al., 2012) are reported. Candidate genes, controlling N, P, and S accumulation associated with previously reported QTLs using SoyBase Genome Browser, are presented.**

| Candidate Genes | Reference Genome | Functional annotation |
|---|---|---|
| Glyma.02G043700/Glyma.02G041900 | Glyma4.0 | Ammonium transporter 2/phosphoserine aminotransferase |
| Glyma.07G045900 | Glyma4.0 | Alanine aminotransferase 2 |
| Glyma.15G015600 | Glyma4.0 | Nodulin MtN21/EamA-like transporter family protein |
| Glyma.17G110701/Glyma.17G106800 | Glyma4.0 | Nodulin-like/ Major Facilitator Superfamily protein/ glutamine-dependent NAD(+) synthetase |
| Glyma.01G033700/Glyma.01G033800/ Glyma.01G033900/Glyma.01G034300 | Glyma4.0 | Nodulin MtN21/EamA-like transporter family/ YLS7-like associated with nitrogen metabolism |
| Glyma.04G058600/Glyma.04G060600/ Glyma.04G073800 | Glyma4.0 | NAD(P)-binding Rossmann-fold superfamily protein/ early nodulin-like protein 15/N-acetyl-l-glutamate kinase |
| Glyma.04G073800/Glyma.04G080700 | Glyma4.0 | Early nodulin-like protein 15/N-acetyl-l-glutamate kinase/ aspartate aminotransferase 3 |
| Glyma.06G055100 | Glyma4.0 | Nitrilase-like protein 1 |
| Glyma.10G030800 | Glyma4.0 | Ammonium transmembrane transporter |
| Glyma.16G041200 | Glyma4.0 | Glutamate dehydrogenase 1 |
| Glyma.17G067200/Glyma.17G067301 | Glyma4.0 | Glutamate receptor/glutamate receptor 2.8-like protein |
| Glyma.04G088000 | Glyma4.0 | Inositol-tetrakisphosphate 1-kinase 2-like isoform X1 |
| Glyma.05G020500 | Glyma4.0 | Probable glycerophosphoryl diester phosphodiesterase 3-like protein |
| Glyma.12G014400/Glyma.12G019600 | Glyma4.0 | Nucleotide-diphospho-sugar transferase super-family protein/putative phosphatidylinositol N-acetylglucosaminyltransferase subunit C-like isoform X1 |
| Glyma.13G004900 | Glyma4.0 | Choline-phosphate cytidylyltransferase |
| Glyma.02G054800 | Glyma4.0 | 1-(5-phosphoribosyl)-5-[(5- phosphoribosylamino) methylideneamino] imidazole-4-carboxamide isomerase |
| Glyma.03G043675 | Glyma4.0 | Protein phosphatase 2C family protein; |
| Glyma.20G011900/Glyma.20G018000 | Glyma4.0 | Purple acid phosphatase 17/Phosphoglucomutase/ phosphomannomutase, |
| Glyma.02G095500 | Glyma4.0 | Sulfate transporter 4.1 |
| Glyma.05G038100 | Glyma4.0 | Iron-sulfur cluster assembly protein |
| Glyma.16G030800 | Glyma4.0 | S-adenosyl-L-methionine-dependent methyltransferases superfamily protein |
| Glyma.02G095500 | Glyma4.0 | Sulfate transporter 4.1 |
| Glyma.04G080300 | Glyma4.0 | Sulfite exporter TauE/SafE family protein |

to leaf minerals and were able to identify, using 200 RILs, 6,366 SNPs markers that covered the whole genome, and 19 QTL and 3 candidate genes associated to leaf-related traits [19]. Previous research, using 92 F$_{5:7}$ (F5-derived) soybean RIL, a cross between MD 96–5722 (MD) and Spencer, and 5,376 SNP markers, were also able to detect QTL related to seed N, P, and S [23]. They identified N QTL *qNIT001* on Chr 16 [23]. They were also identified P QTL *qPHO001* on

Chr16. The S QTL *q SUL001* on Chr16 was also detected [23]. Genomic regions associated with seed N accumulation in soybean at R5, R6, and R7 growth stages were also identified [19]. They used a population of 101 $F_{6:8}$ (F6-derived) RILs, a cross between N87-984–16 × TN93–99 [19]. They detected QTL on Chr 2, 7, 8, 14, 15, 18, and the contribution to the phenotypic variation ranged from 5 to 11.6%.

It must be noted in our research that LOD for N QTL in 2020 was large. This is can be due to the following possible reasons: one is a larger population size (n = 306); unlike the NC site in 2018, which had reduced data due to hurricane damage (n = 187), the IL 2020 trial included the full RIL population. The increased number of data points likely enhanced statistical power, reducing residual variance and inflating LOD values; two, a greater trait variation and data quality, i.e., environmental conditions in Carbondale were more stable, and no external damage affected plant performance. As a result, trait expression was clearer, and stronger marker-trait associations could be detected, especially for seed N, which showed high heritability ($H^2$ = 0.917); third, a narrow confidence intervals and high additive effects: In several cases (e.g., *qN-01*-[IL-2020] and qN-03-[IL-2020]), the trait exhibited strong additive effects combined with tightly linked markers, further contributing to elevated LOD values.

For P, other researchers worked on genomic regions for other minerals such as P. For example, using 184 RILs, a cross between Kefeng No. 1 and Nanong 1138−2 soybean varieties, others [16] were able to identify QTL for P deficiency tolerance in leaves, roots, and shoots, and identified seven QTL associated with weight of fresh shoot, P contents in leaf and in root. Other also identified P QTL associated with shoot P accumulation [20–22]. The low P content heritability has four important aspects: the first aspect is the biological and agricultural relevance, i.e., seed P content remains a biologically and nutritionally important trait in soybean, especially due to its connection with seed vigor, germination, and nutritional quality. Even when heritability is low, understanding the genetic basis of this trait is critical for identifying genotypes or genomic regions responsive to environmental P variability; the second aspect is the precedent in literature, i.e., the low heritability for P-related traits in soybean has been reported previously, especially under single-environment studies or limited replication. For instance, Li et al. (2005) [16] and Zhang et al. (2016) [20] both reported variable P heritability depending on tissue type, environment, and developmental stage. These studies still identified meaningful QTL by using multi-location or repeated trials; similar to our approach; the third aspect is the detection of stable QTL, i.e., despite low overall heritability, we were able to identify several statistically significant QTL for seed P content in both environments (Table 3), with LOD values ranging from 2.5 to 9.4 and phenotypic variance explained ($R^2$) up to 12.17%. This suggests that specific loci still contribute consistently across environments, even if the overall trait expression is environmentally labile; the fourth aspect is the justification through G × E, i.e., the low heritability itself is an important genetic insight. It indicates a potential for strong environmental modulation or G × E interactions, which could be exploited through genotype-specific agronomic strategies or P management practices. Moreover, inclusion of this trait helps distinguish which QTL are environment-stable versus those that are more plastic. Our research showed that, except for QTL detected on Chr 16 [23], 11 QTL reported here were not previously identified, therefore, they are novel.

## Candidate genes and gene annotation for N P S

Candidate genes were identified for N, P, and S concentrations in soybean seeds were annotated using SoyBase Genome Browser (glyma.Wm82.gnm4) (Table 4 and Table 5). These annotated genes were located either within the identified QTL interval or vicinity areas. For N, more than 11 genes, including ammonium transporter, nodulin like proteins, glutamine-dependent NAD (+) synthetase, ammonium transmembrane transporters were identified in the functional loci, suggesting the functional association with N metabolism and accumulation and their genetic variants. These candidate genes may play roles in ammonium transport across cellular membranes, reversible transfer of an amino group between alanine and α-ketoglutarate and forming glutamate and root nodules of leguminous plants. The ammonium transporters (AMTs) are important genes involved in ammonium absorption and utilization

in soybeans. The QTL underlying N concentration were identified on chromosome 2 and 10 in this research and explained more than 8% and 42% phenotypic variation, respectively. The candidate genes associated with these two loci, *Glyma.02G043700* and *Glyma.10G030800* were identified in the intervals. The same candidate genes *GmAMT4.4* (*Glyma.02G043700*) and *GmAMT4.6* (*Glyma.10G030800*) were also annotated, and genes expression of *GmAMT* family genes in the soybean plants, using a GWAS, was conducted by others [46]. Both candidate genes were among 16 *GmAM* paralog genes annotated in the study and GmAMT4.6 was confirmed to be related with the circadian rhythms in the transcription analysis [46]. Moreover, there were several candidate genes existing within the loci for the traits of P and S, suggesting the functional significance of these intervals for the nutrients in soybean. The candidate gene inositol-tetrakisphosphate 1-kinase 2-like isoform X1 (*Glyma.04G088000*) was annotated in Soybase involved in the metabolism of inositol phosphates as the signaling molecules. Protein phosphatase 2C family gene (*Glyma.03G043675*) appeared to be involved in the regulation of crucial steps in the cells by catalyzing the removal of phosphate groups from proteins. The candidate gene purple acid phosphatase 17 (*Glyma.20G011900*) under the QTL (*qP-03*) on chromosome 20 is involved in phosphate metabolism. The QTL (*qP-03*) explained more than 5% phenotypic variation. Zhu et al. (2020) [47] studied the dynamic changes of intracellular (leaf and root) and root-associated APase (purple acid phosphatase) activity under both Pi-sufficient and Pi-deficient conditions for a total of 38 purple acid phosphatase (*GmPAP*) members identified in the soybean genome. The relative expression levels of *GmPAP17c* (*Glyma.20G011900*) and other *GmPAP*s in the leaves and roots were analyzed at two P levels using qRT-PCR analysis. However, the transcript abundance of the *GmPAP17c* was detected at both P levels in the leaves, but not in the roots [47]. There were several candidate genes identified for the sulfur content within the QTL intervals. The sulfate transporter 4.1 (*Glyma.02G095500*) underneath (*qS-01*) is involved in the transport of sulfate ions across cell membranes. The iron-sulfur cluster assembly protein (Gm05_3273418) is involved in electron transfer reactions and other redox processes.

It is clear, based on the above discussion, and except of seed N, P, and S QTL, detected on Chr 16 [23], no QTL of seed N, P, and S were previously detected in mature seed for the accumulation of these nutrients. Therefore, our research showed that 11 QTL reported here were not previously identified, therefore, they are novel.

## Conclusions

In this research a total of 12 QTL were identified for seed N, P, and S at the complete physiological seed maturity (R8), and among which, 11 QTL were novel. Seed N, P, and S QTL on Chr 16 were previously identified by others. The low heritability for seed P QTL could be due to complex quantitative trait, gene-to-gene interactions, and significant gene by environment interactions with P. This suggests the limitation use of P QTL in the breeding selection, but to detect significant QTL with high heritability, a large number of RILs across locations and across years may be required. Therefore, further research for P QTL is needed before final conclusions are made. QTL and molecular markers discovered in this research will help breeders for selection for optimum mineral nutritional qualities; for physiologists to advance our knowledge in the physiology and genetics of seed mineral accumulation, and for molecular biologists with new knowledge on seed N, P, and S candidate genes and their possible use in genetic engineering.

## Supporting information

**S1 File. Genetic background of the parent used in crosses and mapping; relevant information was collected from Soybase using the below link.**
(PDF)

**S2 File. QTL and genetic map for 2018.**
(XLSX)

**S3 File. QTL and genetic map for 2020.**
(XLSX)

**S1 Fig. Chromosome 1 and parameters associated with the quantitative trait loci (QTL) for seed N, P, and S in 'Forrest' by 'Williams 82' recombinant inbred soybean lines (RILs) population. A total of 5405 single nucleotides polymorphism (SNP) markers using Infinium NP6K BeadChips. A total 2075 polymorphic SNPs were mapped on the 20 soybean chromosomes**.
(TIF)

**S2 Fig. Chromosome 2 and parameters associated with the quantitative trait loci (QTL) for seed N, P, and S in 'Forrest' by 'Williams 82' recombinant inbred soybean lines (RILs) population. A total of 5405 single nucleotides polymorphism (SNP) markers using Infinium NP6K BeadChips. A total 2075 polymorphic SNPs were mapped on the 20 soybean chromosomes**.
(TIF)

**S3 Fig. Chromosome 3 and parameters associated with the quantitative trait loci (QTL) for seed N, P, and S in 'Forrest' by 'Williams 82' recombinant inbred soybean lines (RILs) population. A total of 5405 single nucleotides polymorphism (SNP) markers using Infinium NP6K BeadChips. A total 2075 polymorphic SNPs were mapped on the 20 soybean chromosomes**.
(TIF)

**S4 Fig. Chromosome 4 and parameters associated with the quantitative trait loci (QTL) for seed N, P, and S in 'Forrest' by 'Williams 82' recombinant inbred soybean lines (RILs) population. A total of 5405 single nucleotides polymorphism (SNP) markers using Infinium NP6K BeadChips. A total 2075 polymorphic SNPs were mapped on the 20 soybean chromosomes**.
(TIF)

**S5 Fig. Chromosome 5 and parameters associated with the quantitative trait loci (QTL) for seed N, P, and S in 'Forrest' by 'Williams 82' recombinant inbred soybean lines (RILs) population. A total of 5405 single nucleotides polymorphism (SNP) markers using Infinium NP6K BeadChips. A total 2075 polymorphic SNPs were mapped on the 20 soybean chromosomes**.
(TIF)

**S6 Fig. Chromosome 6 and parameters associated with the quantitative trait loci (QTL) for seed N, P, and S in 'Forrest' by 'Williams 82' recombinant inbred soybean lines (RILs) population. A total of 5405 single nucleotides polymorphism (SNP) markers using Infinium NP6K BeadChips. A total 2075 polymorphic SNPs were mapped on the 20 soybean chromosomes**.
(TIF)

**S7 Fig. Chromosome 7 and parameters associated with the quantitative trait loci (QTL) for seed N, P, and S in 'Forrest' by 'Williams 82' recombinant inbred soybean lines (RILs) population. A total of 5405 single nucleotides polymorphism (SNP) markers using Infinium NP6K BeadChips. A total 2075 polymorphic SNPs were mapped on the 20 soybean chromosomes**.
(TIF)

**S8 Fig. Chromosome 10 and parameters associated with the quantitative trait loci (QTL) for seed N, P, and S in 'Forrest' by 'Williams 82' recombinant inbred soybean lines (RILs) population. A total of 5405 single nucleotides**

polymorphism (SNP) markers using Infinium NP6K BeadChips. A total 2075 polymorphic SNPs were mapped on the 20 soybean chromosomes.
(TIF)

**S9 Fig. Chromosome 12 and parameters associated with the quantitative trait loci (QTL) for seed N, P, and S in 'Forrest' by 'Williams 82' recombinant inbred soybean lines (RILs) population. A total of 5405 single nucleotides polymorphism (SNP) markers using Infinium NP6K BeadChips. A total 2075 polymorphic SNPs were mapped on the 20 soybean chromosomes.**
(TIF)

**S10 Fig. Chromosome 13 and parameters associated with the quantitative trait loci (QTL) for seed N, P, and S in 'Forrest' by 'Williams 82' recombinant inbred soybean lines (RILs) population. A total of 5405 single nucleotides polymorphism (SNP) markers using Infinium NP6K BeadChips. A total 2075 polymorphic SNPs were mapped on the 20 soybean chromosomes.**
(TIF)

**S11 Fig. Chromosome 15 and parameters associated with the quantitative trait loci (QTL) for seed N, P, and S in 'Forrest' by 'Williams 82' recombinant inbred soybean lines (RILs) population. A total of 5405 single nucleotides polymorphism (SNP) markers using Infinium NP6K BeadChips. A total 2075 polymorphic SNPs were mapped on the 20 soybean chromosomes.**
(TIF)

**S12 Fig. Chromosome 16 and parameters associated with the quantitative trait loci (QTL) for seed N, P, and S in 'Forrest' by 'Williams 82' recombinant inbred soybean lines (RILs) population. A total of 5405 single nucleotides polymorphism (SNP) markers using Infinium NP6K BeadChips. A total 2075 polymorphic SNPs were mapped on the 20 soybean chromosomes.**
(TIF)

**S13 Fig. Chromosome 17 and parameters associated with the quantitative trait loci (QTL) for seed N, P, and S in 'Forrest' by 'Williams 82' recombinant inbred soybean lines (RILs) population. A total of 5405 single nucleotides polymorphism (SNP) markers using Infinium NP6K BeadChips. A total 2075 polymorphic SNPs were mapped on the 20 soybean chromosomes.**
(TIF)

**S14 Fig. Chromosome 19 and parameters associated with the quantitative trait loci (QTL) for seed N, P, and S in 'Forrest' by 'Williams 82' recombinant inbred soybean lines (RILs) population. A total of 5405 single nucleotides polymorphism (SNP) markers using Infinium NP6K BeadChips. A total 2075 polymorphic SNPs were mapped on the 20 soybean chromosomes.**
(TIF)

**S15 Fig. Chromosome 20 and parameters associated with the quantitative trait loci (QTL) for seed N, P, and S in 'Forrest' by 'Williams 82' recombinant inbred soybean lines (RILs) population. A total of 5405 single nucleotides polymorphism (SNP) markers using Infinium NP6K BeadChips. A total 2075 polymorphic SNPs were mapped on the 20 soybean chromosomes.**
(TIF)

**S16 Fig. Example of some significant QTL as indicated by their logarithms of adds (LOD) for N (top) on Chromosome 1; P (middle) on chromosome 3; N (bottom) on chromosome 4.**
(TIF)

## Acknowledgments

We would like to thank Tri D. Vuong for his technical support on the SoySNP6K Infinium genotyping and linkage map construction. Technical support provided by Sandra Mosley and Rama Gadi is also appreciated. Mention of trade names or commercial products in this publication is solely for the purpose of providing specific information and does not imply recommendation or endorsement by the United States Department of Agriculture (USDA). The findings and conclusions in this publication are those of the authors and should not be construed to represent any official USDA or U.S. Government determination or policy. USDA is an equal opportunity provider and employer.

## Author contributions

**Conceptualization:** Nacer Bellaloui, Jiazheng Yuan, Dounya Knizia, Qijian Song, Frances Betts, Teresa Register, Earl Williams, Naoufal Lakhssassi, Hamid Mazouz, Henry T. Nguyen, Khalid Meksem, Alemu Mengistu, My Abdelmajid Kassem.

**Data curation:** Nacer Bellaloui, Jiazheng Yuan, Henry T. Nguyen, Khalid Meksem, My Abdelmajid Kassem.

**Formal analysis:** Nacer Bellaloui, Jiazheng Yuan, Dounya Knizia, Khalid Meksem, My Abdelmajid Kassem.

**Funding acquisition:** Nacer Bellaloui, Henry T. Nguyen, Khalid Meksem, My Abdelmajid Kassem.

**Investigation:** Nacer Bellaloui, Jiazheng Yuan, Dounya Knizia, Qijian Song, Frances Betts, Teresa Register, Earl Williams, Naoufal Lakhssassi, Hamid Mazouz, Henry T. Nguyen, Khalid Meksem, Alemu Mengistu, My Abdelmajid Kassem.

**Methodology:** Nacer Bellaloui, Jiazheng Yuan, Dounya Knizia, Qijian Song, Frances Betts, Teresa Register, Earl Williams, Naoufal Lakhssassi, Hamid Mazouz, Henry T. Nguyen, Khalid Meksem, My Abdelmajid Kassem.

**Project administration:** Nacer Bellaloui, Henry T. Nguyen, Khalid Meksem, Alemu Mengistu, My Abdelmajid Kassem.

**Resources:** Nacer Bellaloui, Jiazheng Yuan, Qijian Song, Henry T. Nguyen, Khalid Meksem, Alemu Mengistu, My Abdelmajid Kassem.

**Software:** Nacer Bellaloui, Jiazheng Yuan, Dounya Knizia, Khalid Meksem, My Abdelmajid Kassem.

**Supervision:** Nacer Bellaloui, Henry T. Nguyen, Khalid Meksem, My Abdelmajid Kassem.

**Validation:** Nacer Bellaloui, Jiazheng Yuan, Dounya Knizia, Qijian Song, Frances Betts, Teresa Register, Earl Williams, Naoufal Lakhssassi, Hamid Mazouz, Henry T. Nguyen, Khalid Meksem, Alemu Mengistu, My Abdelmajid Kassem.

**Visualization:** Nacer Bellaloui, Jiazheng Yuan, Dounya Knizia, Qijian Song, Frances Betts, Teresa Register, Earl Williams, Naoufal Lakhssassi, Hamid Mazouz, Henry T. Nguyen, Khalid Meksem, Alemu Mengistu, My Abdelmajid Kassem.

**Writing – original draft:** Nacer Bellaloui.

**Writing – review & editing:** Nacer Bellaloui, Jiazheng Yuan, Dounya Knizia, Qijian Song, Frances Betts, Teresa Register, Earl Williams, Naoufal Lakhssassi, Hamid Mazouz, Henry T. Nguyen, Khalid Meksem, Alemu Mengistu, My Abdelmajid Kassem.

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
