## [Decision Letter · Decision Letter 0]

21 Apr 2025

Dear Dr. Bellaloui,

We look forward to receiving your revised manuscript.

Kind regards,

Hao-Xun Chang, Ph.D.

Academic Editor

PLOS ONE

Journal Requirements:

https://www.mdpi.com/2037-0164/15/2/35

In your revision ensure you cite all your sources (including your own works), and quote or rephrase any duplicated text outside the methods section. Further consideration is dependent on these concerns being addressed.

 [This research was funded by the U.S. Department of Agriculture, Agricultural Research Service Project 6066-21220-016-000D, SIUC, UM, and FSU.]. 

Reviewers' comments:

Reviewer's Responses to Questions

**Comments to the Author**

1. Is the manuscript technically sound, and do the data support the conclusions?

Reviewer #1: Yes

Reviewer #2: Yes

2. Has the statistical analysis been performed appropriately and rigorously?

Reviewer #1: Yes

Reviewer #2: Yes

3. Have the authors made all data underlying the findings in their manuscript fully available?

Reviewer #1: Yes

Reviewer #2: Yes

4. Is the manuscript presented in an intelligible fashion and written in standard English?

Reviewer #1: Yes

Reviewer #2: No

Reviewer #1: The trait seed P content has near zero heritability meaning no genotypic variation for the trait. Then all the analysis will be wrong. How to justify your study on this trait. There is some english corrections in the manuscript which needs to attended. There is markings done in the manuscript which needs to be attended. Some discussion is excessively written which needs to addressed before the final acceptance of the research paper.

Reviewer #2: The manuscript aims to identify new genomic regions associated with seed nitrogen, phosphorus, and sulfur accumulation with the Forrest by Williams 82 RIL mapping population. Several references were cited in the manuscript without the background information in the materials and methods, it would be beneficial if more in-depth information was provided throughout the article. While the manuscript is insightful, it requires major revision before it can be published. Some examples are listed below.

Introduction:

A good start for the introduction, more background information should be provided. For example, the introduction should briefly discuss the articles referenced, it would be best to directly reference the authors (Line 81 for example, which author; similarly in Line 97, directly citing the author would make the article easier to follow along) Several editorial recommendations include:

Line 66 Soybean missing scientific name

Line 85 full name of the essential nutrients

Line 88 missing , in 6366

Line 97 F6:8

Line 98 N87-984-16 x TN93-99 => × (cross)

Line 98 Chrs 2, 7, 8, 14, 15, 18, suggest including linkage group after the chromosome numbers

Line 99 others (who?)

QTL first appears in article -> Quantitative trait loci

Line 101 “seven” QTL

Line 102 phosphorus => P

Line 103 F5:7

Materials and Methods

Are there any reasons why this mapping population was selected for the study? Forrest was initially released for soybean cyst nematode resistance whereas William 82 was released for Phytophthora root rot resistance. The information in Line 136 to 141 should be included on the top along with the parental line. It would be best for the authors to provide more background information for the readers to follow along.

The cultivars in the study should include the PI number and provide citation

Line 127 The parental lines should include details of the parental lines such as the year that it was released, which research station developed the cultivar, agronomic traits such as flower color, growth habit, seed composition, maturity groups etc.

Were the experiments conducted on university research farms? More details are needed.

Experimental design and replications are not discussed

Line 139 What type of amino acids?

The manuscript references many literatures but does not provide sufficient details for the readers to follow along. For example, it does not reference the BARCSoySNP6k Chip in the texts.

Inconsistency regarding the units throughout the article, some have space while others do not

Line 182 provide link to SoyBase

Statistical Analysis

Has the author compared different QTL mapping methods such as CIM vs Multiple QTL Mapping?

Tables lack units

Please check if the SNP markers have a ss name in SoyBase.

The LOD score for Carbondale IL location is large, is there any reason behind this? Include linkage group information on the tables.

The cultivar names in the manuscript should be consistent, for example Williams 82 only needs the ‘’ the first time it appears in the texts.

Inconsistency in the way numbers are written and how the data is presented, Line 233 for example which chromosomes, is it seven chromosomes or Chr. 7?

The range for the phenotypic variation in the Carbondale, IL location is rather large, is there a reason for that (7.14 – 78.52%)?

Line 259 qP-02 should be italicized

Line 349 F5:7

The discussion and candidate gene annotation section is well written. However, it does not conduct any studies to validate the QTL.

**Do you want your identity to be public for this peer review?** For information about this choice, including consent withdrawal, please see our Privacy Policy

Reviewer #1: No

Reviewer #2: No

---

## [Author Response · Author response to Decision Letter 1]

7 Jul 2025

Dear Dr. Hao-Xun Chang, Academic Editor, PLOS ONE:

Please receive our revised manuscript entitled “Genomic Regions and Candidate Genes Associated with Seed Nitrogen, Phosphorus, and Sulfur Accumulation Identified in the Soybean ‘Forrest’ by ‘Williams 82’ RIL Population” by Bellaloui et al.

We revised the manuscript and addressed all the reviewers’ comments as instructed, on one-by-one basis, and all are track-changes. We also addressed the comments noted in the attached file (form Reviewer 1) either as suggested or as appropriate. Our responses are below. We stated the reviewer’s comments and responded to them, as shown below. Enormous efforts and time were devoted in revising this manuscript, and we believe the reviewers’ comments improved the quality of the manuscript. We would like to note that “ The funders had no role in study design, data collection and analysis, decision to publish, or preparation of the manuscript."

Thank you and we appreciate your service,

Reviewers comments and Authors’ Responses

Journal Requirements:

Authors' response

All journal requirements were addressed.

Authors’ response

We revised the entire manuscript, and fixed the files’ names for tables, fugues, files, and so on according to the journal requirements.

https://www.mdpi.com/2037-0164/15/2/35

In your revision ensure you cite all your sources (including your own works), and quote or rephrase any duplicated text outside the methods section. Further consideration is dependent on these concerns being addressed.

Authors’ response

We revied the manuscript and made sure there are no similarities, unless the similarities are scientific terms or technical names. In addition, that published manuscript dealt with Fe and Zn, but the current manuscript deals with N, S, and P.

[This research was funded by the U.S. Department of Agriculture, Agricultural Research Service Project 6066-21220-016-000D, SIUC, UM, and FSU.].

Please state what role the funders took in the study. If the funders had no role, please state: ""The funders had no role in study design, data collection and analysis, decision to publish, or preparation of the manuscript."

We include the appropriate statement in the manuscript as instructed “

Authors’ response

“The funders had no role in study design, data collection and analysis, decision to publish, or preparation of the manuscript."

Authors’ response

We included it in the cover letter as well, as below:

“ The funders had no role in study design, data collection and analysis, decision to publish, or preparation of the manuscript."

N/A

Reviewers' comments:

Reviewer's Responses to Questions

Comments to the Author

1. Is the manuscript technically sound, and do the data support the conclusions?

Reviewer #1: Yes

Reviewer #2: Yes

Authors’ response

Thank you for the comment.

2. Has the statistical analysis been performed appropriately and rigorously?

Reviewer #1: Yes

Reviewer #2: Yes

Authors’ response

Thank you for the comment.

3. Have the authors made all data underlying the findings in their manuscript fully available?

Reviewer #1: Yes

Reviewer #2: Yes

Authors’ response

Thank you for the comment.

4. Is the manuscript presented in an intelligible fashion and written in standard English?

Reviewer #1: Yes

Reviewer #2: No

Authors’ response

We revised the manuscript and made sure the presentation and English writing is sound.

5. Review Comments to the Author

Reviewer #1: The trait seed P content has near zero heritability meaning no genotypic variation for the trait. Then all the analysis will be wrong. How to justify your study on this trait. There is some English corrections in the manuscript which needs to attended. There is markings done in the manuscript which needs to be attended. Some discussion is excessively written which needs to addressed before the final acceptance of the research paper.

Our response

We appreciate the reviewer’s important observation regarding the low broad-sense heritability (H² ≈ 0) for seed phosphorus (P) content. Indeed, low heritability suggests that the phenotypic variation is primarily governed by environmental effects and/or strong genotype-by-environment (G×E) interactions, which complicates genetic dissection. However, we respectfully argue that this does not invalidate the analysis, for the several below reasons (we further explained the below concepts in the Discussion section for the benefit for the reader, and all are track-changed).

1. Biological and Agricultural Relevance

Seed P content remains a biologically and nutritionally important trait in soybean, especially due to its connection with seed vigor, germination, and nutritional quality. Even when heritability is low, understanding the genetic basis of this trait is critical for identifying genotypes or genomic regions responsive to environmental P variability.

2. Precedent in Literature

Low heritability for P-related traits in soybean has been reported previously, especially under single-environment studies or limited replication. For instance, Li et al. (2005) and Zhang et al. (2016) both reported variable P heritability depending on tissue type, environment, and developmental stage. These studies still identified meaningful QTL by using multi-location or repeated trials; similar to our approach.

3. Detection of Stable QTL

Despite low overall heritability, we were able to identify several statistically significant QTL for seed P content in both environments (see Table 3), with LOD values ranging from 2.5 to 9.4 and phenotypic variance explained (R²) up to 12.17%. This suggests that specific loci still contribute consistently across environments, even if the overall trait expression is environmentally labile.

4. Justification Through G×E

The low heritability itself is an important genetic insight. It indicates a potential for strong environmental modulation or G×E interactions, which could be exploited through genotype-specific agronomic strategies or P management practices. Moreover, inclusion of this trait helps distinguish which QTL are environment-stable versus those that are more plastic.

References used above are below, and they are included in the manuscript.

Li, Y.D., Wang, Y.J., Tong, Y.P., Gao, J.G., Zhang, J.S., & Chen, S.Y. (2005). QTL mapping of phosphorus deficiency tolerance in soybean (Glycine max L. Merr.). Euphytica, 142, 137–142. https://doi.org/10.1007/s10681-005-1047-2.

Zhang, D., Li, H., Wang, J., Zhang, H., Hu, Z., Chu, S., Lv, H., & Yu, D. (2016). High-density genetic mapping identifies new major loci for tolerance to low-phosphorus stress in soybean. Frontiers in Plant Science, 7, 372. https://doi.org/10.3389/fpls.2016.00372.

Reviewer #2: The manuscript aims to identify new genomic regions associated with seed nitrogen, phosphorus, and sulfur accumulation with the Forrest by Williams 82 RIL mapping population. Several references were cited in the manuscript without the background information in the materials and methods, it would be beneficial if more in-depth information was provided throughout the article. While the manuscript is insightful, it requires major revision before it can be published. Some examples are listed below.

Authors response:

We revised the entire manuscript for better comprehension and further in-depth information, and we paid attention to the cited references in Materials and methods for further details and backgrounds. All edits are track-changes.

Introduction:

A good start for the introduction, more background information should be provided. For example, the introduction should briefly discuss the articles referenced, it would be best to directly reference the authors (Line 81 for example, which author; similarly in Line 97, directly citing the author would make the article easier to follow along) Several editorial recommendations include:

Authors response:

The introduction was revised and adjusted as needed for better clarification. Citations were directly added for better clarification. We worry that adding more information will be in conflict with the suggestion of Reviewer 1 as Reviewer 1 indicated that some discussion is excessively written.

Line 66 Soybean missing scientific name

Line 85 full name of the essential nutrients

Line 88 missing , in 6366

Line 97 F6:8

Line 98 N87-984-16 x TN93-99 => × (cross)

Line 98 Chrs 2, 7, 8, 14, 15, 18, suggest including linkage group after the chromosome numbers

Line 99 others (who?)

QTL first appears in article -> Quantitative trait loci

Line 101 “seven” QTL

Line 102 phosphorus => P

Line 103 F5:7

Authors response:

The above suggested comments were addressed as suggested. We included the linkage group in all tables.

Materials and Methods

Reviewer’s comment

Are there any reasons why this mapping population was selected for the study? Forrest was initially released for soybean cyst nematode resistance whereas William 82 was released for Phytophthora root rot resistance. The information in Line 136 to 141 should be included on the top along with the parental line. It would be best for the authors to provide more background information for the readers to follow along.

Authors response:

We moved the information to the top along with the parental lines, as suggested. We included detailed about Forrest and Williams 82 in Materials and methods; we included a Figure (Fig 1) and supplementary information/file (S1_File) about the background of the parent and crosses for the benefit of the reader.

Reviewer’s comment

The cultivars in the study should include the PI number and provide citation

Line 127 The parental lines should include details of the parental lines such as the year that it was released, which research station developed the cultivar, agronomic traits such as flower color, growth habit, seed composition, maturity groups etc.

Authors response

We included a figure (Fig 1) and supplementary information/file (S1_File) about the background of the parent. We also detailed their background in the Materials and methods ( we included their PI, their crosses, Fig 1; S1_File) and more for the benefit of the reader. All are included in the text and all are track-changed.

Reviewer’s comment

Were the experiments conducted on university research farms? More details are needed.

Authors response

Thank you for the suggestion. Yes, the experiments were conducted on university research farms, as described in K (2021) [27]. We have updated the manuscript with more details and to clarify the experimental sites, including affiliations and research station details, to ensure full transparency and reproducibility. All new edits are track-changed.

Reviewer’s comment

Experimental design and replications are not discussed

Authors’ response

We thank the reviewer for pointing this out. The experimental design and replication details were originally described in Knizia et al. (2021), but we further included in the manuscript further details for clarity and completeness. We have revised the Materials and methods section to include a full description of the field design, replication, and environmental factors impacting data collection.

Reviewers’ response

Line 139 What type of amino acids?

Authors’ response

Amino acids included threonine, serine, proline, glycine, alanine, cysteine, valine, methionine, phenylalanine, lysine, tryptophan. We included them in the text, and all are track-changed.

Reviewer’s comments

The manuscript references many literatures but does not provide sufficient details for the readers to follow along. For example, it does not reference the BARCSoySNP6k Chip in the texts.

Inconsistency regarding the units throughout the article, some have space while others do not

Line 182 provide link to SoyBase

Authors’ response

-Thank you for the comment. We adjusted the text and provided further information and a citation and reference, below, for better clarity for the reader.

Soybean BARCSoySNP6K: An assay for soybean genetics and breeding research. The Plant Journal (2020) 104, 800–811 doi: 10.1111/tpj.14960.

-We provided the link

Reviewer’s comment

Statistical Analysis

Has the author compared different QTL mapping methods such as CIM vs Multiple QTL Mapping?

Authors’ response

No, we did not, and our reasons for this are below, and also some clarification was added to the Material and methods:

We appreciate the reviewer’s question regarding the QTL mapping approach. In this study, we used Composite Interval Mapping (CIM) implemented in WinQTL Cartographer 2.5, which is a widely accepted and robust method for detecting QTL with moderate to high power and resolution.

We acknowledge that alternative methods such as Multiple QTL Mapping (MQM) or Inclusive Composite Interval Mapping (ICIM) may offer complementary insights, particularly in separating closely linked QTL or

---

## [Decision Letter · Decision Letter 1]

13 Aug 2025

Genomic Regions and Candidate Genes Associated with Seed Nitrogen, Phosphorus, and Sulfur Accumulation Identified in the Soybean ‘Forrest’ by ‘Williams 82’ RIL Population

PONE-D-24-39665R1

Dear Dr. Bellaloui,

We’re pleased to inform you that your manuscript has been judged scientifically suitable for publication and will be formally accepted for publication once it meets all outstanding technical requirements. Please note that the reviewer 2 pointed out a few minor edits, and I would suggest to consider these revisions in the galley proof stage.

Kind regards,

Hao-Xun Chang, Ph.D.

Academic Editor

PLOS ONE

Reviewers' comments:

Reviewer's Responses to Questions

**Comments to the Author**

Reviewer #2: All comments have been addressed

2. Is the manuscript technically sound, and do the data support the conclusions?

Reviewer #2: Yes

3. Has the statistical analysis been performed appropriately and rigorously?

Reviewer #2: Yes

4. Have the authors made all data underlying the findings in their manuscript fully available?

Reviewer #2: Yes

5. Is the manuscript presented in an intelligible fashion and written in standard English?

Reviewer #2: Yes

Reviewer #2: The authors have addressed the reviewer’s comment, and the manuscript has improved significantly. The newly revised introduction and material and method provide a better understanding of the study. The results and conclusions are consistent with the research question. Several minor edits should be addressed:

Line 156 Forrest does not need ‘’, this should be consistent throughout the text with the cultivars.

Line 340 “nine” chromosomes, “seven” chr, and “nine” QTL

**Do you want your identity to be public for this peer review?** For information about this choice, including consent withdrawal, please see our Privacy Policy

Reviewer #2: No

---

## [Editor Report · Acceptance letter]

PONE-D-24-39665R1

PLOS ONE

Dear Dr. Bellaloui,

I'm pleased to inform you that your manuscript has been deemed suitable for publication in PLOS ONE. Congratulations! Your manuscript is now being handed over to our production team.

Kind regards,

on behalf of

Dr. Hao-Xun Chang

Academic Editor

PLOS ONE